# Genotypic and Phenotypic Characterization of *Escherichia coli* Isolates Recovered from the Uterus of Mares with Fertility Problems

**DOI:** 10.3390/ani13101639

**Published:** 2023-05-14

**Authors:** Francesca Paola Nocera, Linda Maurizi, Angelo Masullo, Mauro Nicoletti, Antonietta Lucia Conte, Francesca Brunetti, Luisa De Martino, Carlo Zagaglia, Catia Longhi

**Affiliations:** 1Department of Veterinary Medicine and Animal Production, University of Naples ‘Federico II’, Via F. Delpino 1, 80137 Naples, Italy; 2Department of Public Health and Infectious Diseases, Sapienza University of Rome, 00185 Rome, Italy

**Keywords:** *Escherichia coli* (*E. coli*), mares, phylogenetic group, biofilm formation, multidrug-resistant strains

## Abstract

**Simple Summary:**

Endometritis, one of the leading causes of mares’ infertility and likely originates from bacteria inhabiting the vaginal microflora which ascend the genitourinary tract. Urogenital infections are caused by opportunistic or commensal microorganisms, and among them *Escherichia coli* strains have frequently been isolated and associated with mare infertility. Herein, we report the phenotypic and genotypic characterization of 24 *E. coli* strains isolated from the uterus of 24 independent mares which had been examined by an equine veterinarian because of fertility problems. Antimicrobial-resistance, biofilm production, the ability to adhere and invade HeLa cells as well as the carriage of specific *E. coli* associated virulence genes were investigated. Particular attention was given to the profiles of *E. coli* strains isolated by direct-plating and those that required enrichment in broth before seeding on the plate.

**Abstract:**

*Escherichia coli* is the bacterial pathogen most frequently associated with mare infertility. Here, we characterized 24 *E. coli* strains isolated from mares which presented signs of endometritis and infertility from a genotypic and phenotypic point of view. The majority of the isolates belonged to phylogenetic group B1 (9/24, 37.5%). Regarding antibiotic resistance profiles, 10 out of 24 (41.7%) were multidrug-resistant (MDR). Moreover, 17 out of 24 (70.8%) were strong or moderate biofilm producers, and of these eight were MDR strains. Interestingly, 21 out of 24 (87.5%) *E. coli* strains were phenotypically resistant to ampicillin and 10 of them were also resistant to amoxicillin with clavulanic acid. Regarding the presence of selected virulence factors, 50% of the examined strains carried at least three of them, with *fimH* detected in all strains, and followed by *kpsMTII* (11/24, 45.9%). No strain was able to invade HeLa cell monolayers. No relevant differences for all the investigated characteristics were shown by strains that grew directly on plates *versus* strains requiring the broth-enrichment step before growing on solid media. In conclusion, this work provides new insight into *E. coli* strains associated with mares’ infertility. These results broaden the knowledge of *E. coli* and, consequently, add useful information to improve prevention strategies and therapeutic treatments contributing to a significant increase in the pregnancy rate in mares.

## 1. Introduction

Infertility or hypofertility in horses is defined as the inability of a horse to conceive. Many factors can affect the horse’s ability to produce offspring including not being in heat during insemination; silent heat; poor nutrition; poor reproductive anatomy; cysts, growths or scars along the reproductive tract that prevent sperm reaching the egg; genetic abnormalities, such as the incidence of chromosomal abnormalities and infections causing endometritis [1].

Regarding bacterial infections and infertility in mares, endometritis is an important cause considering that from 25 to 60% of barren mares present signs of endometritis. Induction of inflammation, epithelial adherence, resistance to phagocytosis and viscosity of secretions vary greatly between pathogens and play an important role in influencing fertility [2]. Furthermore, subclinical endometritis has become a more important reason for subfertility in the mare. For these reasons, preventive antibiotic treatments are highly recommended for a successful breeding outcome. In addition, bacterial infections have been frequently associated with equine fertility and multiple-antibiotic-resistant *Escherichia coli* strains have been described as the principal cause of infection worldwide [3,4,5,6].

Despite its ubiquity as a commensal, *E. coli* is also an important pathogen of humans and horses, being responsible for intestinal as well as extraintestinal infections [7,8,9]. Furthermore, domestic and wild animals can act as reservoirs for human extraintestinal pathogenic *E. coli* [10]. *E. coli* strains which cause infections in tissues other than the intestinal tract are called ExPECs. ExPECs, including different sequence types, are the causative agents of important infections such as urinary tract infections, peritonitis, pneumonia, meningitis, and septicemia; and it seems that these *E. coli* strains cannot be clearly distinguished from intestinal *E. coli* that may serve as the primary source of the *E. coli* extraintestinal infections [11].

The infections caused by antimicrobial-resistant pathogens in horses involve the correct use of antimicrobials balanced with the requirement to treat the presenting clinical condition [12]. Mares are prone to develop chronic infections, which might be due to bacterial biofilms conferring the ability to evade the immune system to microorganisms and to resist antimicrobial therapy [13]. Furthermore, the bacterial pathogens often present essential pathogenicity traits, including adhesion and invasion ability to host cells. The bacterial ability to invade host cells is a pathogenic mechanism which can have serious effects on the establishment, persistence, severity and propagation of infections, as well as the motility mediated by flagella and toxins [14]. 

Phylogenetic studies have shown that *E. coli* belongs to four main phylogenetic groups specifically A, B1, B2, and D. The virulent extra-intestinal *E. coli* strains predominantly belong to group B2 and less extensively to group D, while the commensal strains, considered less virulent, belong to groups A or B1 [15,16].

To the best of our knowledge, there are not many studies on the correlation between antimicrobial resistance, biofilm production, virulence determinants and phylogeny of *E. coli* isolates from mares suffering from endometritis-associated fertility problems. Hence, this study was designed to assess the virulence potential of equine *E. coli* strains in Southern Italy. Furthermore, another objective of this study was to compare the results obtained from *E. coli* strains isolated after direct-plating and those that needed the use of a broth-enrichment step before plating using the same media.

## 2. Materials and Methods

### 2.1. Ethical Approval

The bacteriological examinations were performed for clinical purposes, thus Ethics Committee approval was not required. All the *E. coli* strains, herein used, are part of the bacteriotheque of the Microbiological Diagnostic Laboratory of the Department of Veterinary Medicine and Animal Production, University of Naples “Federico II” (Italy), where the strains were collected for routine veterinary investigations, and were obtained from uterine swabs of mares diagnosed with suspected bacterial endometriosis by clinical equine veterinarians.

### 2.2. Sample Collection

In 2019, 24 *E. coli* strains were collected from routine bacteriological examinations of equine samples sent to the Microbiological Diagnostic Laboratory of the Department of Veterinary Medicine and Animal Production, University of Naples “Federico II” (Italy). All strains originated from uterine swabs collected from mares diagnosed with suspected bacterial endometritis by clinical equine veterinarians for their history of repetitive infertility, such as being barren in the previous season, abortion/resorption, repeated breeding during the season, presence of uterine fluid, vaginitis and vaginal discharge. After sampling, uterine swabs were transported at 4 °C to the microbiology laboratory in a transportation box.

### 2.3. Cell Line

The human cervical cancer cell line, HeLa (ATCC CCL-2), was used for the adhesion and invasion assays. HeLa cells were grown in Dulbecco Modified Eagle Medium (DMEM—Gibco^®^, Grand Island, NY, USA), that was supplemented with 10% fetal bovine serum (FBS—Gibco^®^) and 1% antimicrobial solution (penicillin 100 IU/mL, streptomycin 100 μg/mL and amphotericin B 2.5 μg/mL—Gibco^®^). Cells were maintained at 37 °C in a humidified atmosphere with 5% CO_2_.

### 2.4. E. coli Isolation and Identification

Once uterine swabs arrived at the laboratory, they were cultured and streaked in parallel on different selective agar plates and in broth-enrichment Brain Heart infusion (BHI) (Liofilchem Srl, Teramo, Italy), then incubated aerobically at 37 °C for 24 h, as previously described by Nocera et al. [17]. After an overnight incubation, turbid broths were subcultured on the same agar plates. *E. coli* strain isolation was carried out by using plates of MacConkey (MCA) agar (Liofilchem Srl, Teramo, Italy), a selective and differential medium for gram-negative bacteria. Once the bacteria revealed themselves on MCA plates, suspected lactose fermenting *E. coli* colonies were firstly subjected to standard, rapid screening techniques: colony morphology on MCA and tryptic soy blood agar (TSA) plates (Liofilchem Srl, Teramo, Italy), and cellular morphology by Gram’s staining method as well as catalase and oxidase tests. Identification of bacterial strains was performed using matrix-assisted laser desorption ionization time-of-flight mass spectrometry (MALDI–TOF MS) (Bruker Daltonics GmbH, Bremen, Germany), according to manufacturer’s guidelines. Specifically, a single colony, picked from each pure culture, was diluted on a MALDI–TOF MS target plate. Subsequently, 1 µL of matrix solution was deposited onto each sample spot and allowed to air dry at room temperature. The target was then introduced into the MALDI–TOF MS for automated measurement and data interpretation. For identification, all samples were run in duplicates. *E. coli* ATCC^®^ 25922TM was included as a quality control microorganism.

### 2.5. Hemolysis 

Hemolysin production was assessed with the use of plates containing 5% defibrinated sheep blood (Oxoid Ltd., Basingstoke, UK). The plates were examined after up to 48 h of incubation at 37 °C for the presence of a hemolysis area around colonies [18].

### 2.6. Evaluation of E. coli Antimicrobial Susceptibility Profiles

The antimicrobial susceptibility profiles of the isolated strains were determined by the disk diffusion method on Mueller–Hinton agar plates (Oxoid Ltd., Basingstoke, UK). The inhibitory zone diameters obtained around the antibiotic disks were measured after incubation for 24 h at 37 °C and evaluated according to the Clinical and Laboratory Standards Institute [19] and to the European Committee on Antimicrobial Susceptibility Testing [20], allowing the classification of the strains as susceptible (S), intermediate (I) and resistant (R). Fifteen commercial antimicrobial disks purchased from Liofilchem Srl (Teramo, Italy) were tested against the isolated strains to detect trends of resistance associated with them. Antimicrobial agents were: amoxycillin/clavulanic acid (AUG, 20/10 μg), amikacin (AK, 30 μg), ampicillin (AMP, 10 μg), ceftiofur (EFT, 30 μg), ceftriaxone (CRO, 30 μg), ceftazidime (CAZ 30 μg), enrofloxacin (ENR, 5 μg), gentamicin (CN, 10 μg), kanamycin (K, 30 μg), meropenem (MRP, 10 μg), norfloxacin (NOR, 10 μg), ofloxacin (OFX, 5 μg), sulfamethoxazole–trimethoprim (SXT, 25 μg), tetracycline (TE, 30 μg), oxytetracycline (T, 30 μg). Since the tested antimicrobials belonged to seven different antimicrobial classes, strains which resulted in being resistant to at least one agent in three or more antimicrobial classes were defined as multidrug-resistant strains in accordance with Magiorakos et al. [21]. The tested antimicrobials and the respective classes are reported in Table 1. All *E. coli* strains were stored at −20 °C using Microbank™, a ready-to-use system for storage and retrieval of bacterial cultures (Pro Lab Diagnostics, Round Rock, TX, USA).

### 2.7. Evaluation of E. coli Biofilm Formation

Biofilm assays were conducted in 96-well polystyrene microplates, inoculating 20 µL of each bacterial strain (1–2 × 10^8^ CFU/mL) in a well filled with 180 μL of Tryptic Soy Broth (TSB). Plates were incubated at 37 °C for 24 h. After the incubation time, microplates were washed twice with phosphate-buffered saline (PBS), allowed to dry, and fixed with methanol (99.8% *v*/*v*) for 15 min. Subsequently, wells were stained for 20 min with crystal violet (Sigma-Aldrich, 1% *w*/*v*), a basic dye that binds negatively charged molecules, then rinsed three times with H_2_O, and eluted with 95% ethanol. The optical density (OD) at 570 nm was measured by a microplate reader (Bio-rad Benchmark, Hercules, CA, USA), and biofilm production was determined as described by Stepanović et al. [22]. Based on the cut-off OD, defined as three standard deviations above the mean OD of the negative control (ODc), strains were classified as non-biofilm producers (OD ≤ ODc), weak biofilm producers ODc < OD ≤ (2 × ODc), moderate biofilm producers (2 × ODc) < OD ≤ (4 × ODc) and strong biofilm producers (4 × ODc) < OD.

### 2.8. Phylotyping

Belonging to a specific phylogroup was assessed by a PCR assay targeting the genes *chuA* (required for heme transport in enterohemorrhagic O157:H7 *E. coli*), *yjaA* (initially identified in the complete genome sequence of *E. coli* K-12, the function of which is unknown) and an anonymous DNA fragment *TspE4.C2*. Based on the obtained amplification pattern *E. coli* strains are assigned to one of the main phylogroups, A, B1, B2 or D according to the scheme published by Clermont et al. [16]. Amplifications were carried out in 25 μL reaction mixtures composed of 1X PCR reaction buffer (Biolabs Inc., New England, UK), 50 ng/μL of whole DNA bacterial extracts, 0.2 mM of dNTPs (Biolabs Inc.), 0.5 μM of primers (Sigma-Aldrich, Milan, Italy) and 1.25 U Taq DNA polymerase (Biolabs Inc., New England, UK). Each cycle consisted of a denaturing step of 30 s at 94 °C, followed by an annealing step of 30 s at 56 °C and an extension step at 72 °C for 30 s. After 30 cycles of PCR, amplicons were separated by 2.0% agarose gel electrophoresis in 45 mM Tris-borate, 1 mM EDTA buffer (pH 8.0) containing 5 μL of Midori Green Advance DNA stain (Nippon Genetics Europe GmbH, Germany). Amplifications were performed in triplicate and their size was determined by comparison with a 100-bp DNA ladder (Biolabs Inc., New England, UK).

### 2.9. Evaluation of E. coli Virulence Genes (VGs)

All *E. coli* strains were assayed for selected virulence genes by multiplex PCR using appropriate primers (Table 2), as already described by Johnson and Stell [23]. Genes associated with adhesin (*fimH*), capsule synthesis (*kpsMTII*), iron acquisition (*fyuA* and *iutA*), toxins (*cnf1*), invasin (*ibeA*) and serum resistance (*traT*) were investigated. Whole DNA bacterial extracts were prepared using a Qiagen DNA extraction kit (Qiagen, Milan, Italy). *E. coli* strains known to encode the assayed virulence traits were included as controls. PCR was carried out in 12.5 μL in 1X reaction buffer (Euroclone, Milan, Italy). Each reaction mixture contained 50 ng/μL of each DNA template, 1.5 mM MgCl_2_, 0.2 mM dNTPs (Biolabs Inc., New England, UK), 0.5 μM of each primer pairs, 1.25 U of Taq DNA polymerase (Euroclone, Milan, Italy). Amplification was carried out for 25 cycles consisting of denaturation (94 °C, 30 s), annealing (63 °C, 30 s), and extension (68 °C, 3 min) and a final extension (72 °C, 10 min). The products of amplification were separated by 2.0% agarose gel electrophoresis. Amplifications were performed in triplicate and the size of amplicons was determined by comparison with a 100-bp DNA ladder (Biolabs Inc., New England, UK). Each positive gene was confirmed with a separate PCR.

### 2.10. Adhesion Assay

For the adhesion assay, HeLa cells, cultured in 24-well plates at a density of 1 × 10^5^ cells/well for 24 h, were infected with each *E. coli* strain at a multiplicity of infection (MOI) of 10 bacteria/cell. To promote bacterial adherence to the cell monolayer, plates were centrifuged twice at 500× *g* for 2.5 min and subsequently incubated for 30 min at 37 °C in 5% CO_2_. After the incubation period, cell monolayers were washed with PBS, lysed with ice-cold 0.1% Triton X-100 and plated onto TSA to determine the vital count on plates. Bacterial adhesion was defined as the percentage of attached bacteria compared with the initial inoculum. Bacteria were considered adhesive when the percentage was ≥0.8%. The adhesiveness values have been divided into: (+) for a value range of 1–9%; (++) between 10–19%; (+++) for values equal to or over 20%.

### 2.11. Invasion Assays

The invasive ability of isolates was determined by the gentamicin protection assay, as described by Longhi et al. [24]. Bacteria were grown overnight in TSB broth at 37 °C, and HeLa cell monolayers, cultured in 24-well plates at a density of 2 × 10^5^ cells/well, were infected with the bacterial suspensions at a MOI of 10 bacteria/cell. Plates were centrifuged as in the adhesion assay and incubated for 1 h at 37 °C. At the end of incubation time, cell monolayers were washed and fresh medium with 100 μg/mL of gentamicin (Sigma) was added to each well and incubation was continued for 1 h. Monolayers were extensively washed with PBS, lysed with a cold solution of 0.1% *v*/*v* Triton X-100 and plated in TSA. Dilutions of cell lysates were plated on TSA agar plates and the number of viable bacteria counted. A strain was considered invasive when the ratio of the number of intracellular bacteria/initial inoculum×100 was ≥0.1%. *E. coli* MG1655 and *E. coli* AIEC LF82 strains were utilized as negative and positive controls, respectively.

### 2.12. Hierarchical Clustering Analysis

Hierarchical clustering analysis was performed using R programming language v. 4.1.2 (https://www.R-project.org/) and its cluster package [25]. To measure the similarity between the strains, a dissimilarity matrix was generated using the Gower distance index and evaluated the following features: the phylogenetic group, the ability to form biofilm, the identified VF genes, the AMR profile, and whether the strain underwent a broth-enrichment step before isolation. Clusters were then generated with an average-linkage agglomerative approach and statistically validated with the permutational multiple analysis of variance (PERMANOVA) test, performing 1000 permutations. A *p*-value ≤ 0.05 was considered statistically significant. The dendrogram and the features of the strains were displayed together using the R package ggtree [26,27,28].

### 2.13. Statistical Analysis

Statistical analysis was carried out with R v. 4.1.2, performing non-parametric univariate statistical tests (Chi-squared and Fisher’s exact test) on categorical variables. When necessary, *p*-values were corrected via the Benjamin–Hochberg false discovery rate (FDR) procedure to account for multiple hypothesis testing. A *p*-value ≤ 0.05 was considered statistically significant.

## 3. Results

### 3.1. E. coli Isolation and Identification

From examined equine uterine swabs of the year 2019, 24 *E. coli* strains were recovered, and among them, 11 strains (45.8%) grew directly on MCA plates, whilst the remaining 13 strains (54.1%) required the broth-enrichment step before growing on MCA media. Hemolytic phenotypes on blood agar were detected only in two of the collected *E. coli* strains (8.3%). Identification by MALDI TOF–MS gave a log(score) ≥ 2.4 for all isolated strains, indicative of an optimal genus and species-level identification. 

### 3.2. Antimicrobial Resistance Profiles of E. coli Strains

The antimicrobial resistance profiles showed high resistance to ampicillin (95.5%), amoxicillin–clavulanic acid and tetracycline (41.7%), whereas the most efficient antimicrobial was amikacin (100% of cultures), followed by enrofloxacin (95.8%), ofloxacin (91.7%), ceftiofur (86.9%) and gentamicin (75.0%). In addition, multidrug resistance profiles were detected in 41.7% (10/24) of the isolated *E. coli* strains as reported in Table 3, where antimicrobial resistance profiles for each strain were also described. The analysis of the MDR strains revealed no statistically significant difference between the group consisting of strains isolated after direct-plating (5/11 strains) when compared to the *E. coli* group requiring the additional broth-enrichment step before plating (5/13 strains).

### 3.3. In Vitro E. coli Biofilm Formation

Most of the recovered *E. coli* strains were classified as moderate biofilm producers 54% (N = 13/24), 29% (N = 7/24) of them resulted in being weak biofilm producers, and 17% (N = 4/24) of *E. coli* strains were regarded as strong biofilm producers as described in Table 4.

### 3.4. Phylogenetic Group and E. coli Virulence Genes

To determine the phylogenetic distribution of these strains, the genes *chuA*, *yjaA*, and *TspE4.C2* were examined by PCR and classified into four phylogenetic groups, A, B1, B2, and D. The majority of the isolates belonged to phylogenetic group B1 (9/24, 37.5%), followed by group A (7/24, 29.16%), group D (6/24, 25%,) and group B2 (2/24, 8.3%) (Table 5). Interestingly, among the strains obtained after the additional broth-enrichment step, the predominant phylogenetic group was D with five strains, whilst the phylogenetic group B1 was predominant among the strains that were positive by direct-plating (Table 5).

Among the strains which were positive only after enrichment in broth, two of them (ID 3 and ID 14) were characterized using the hemolytic phenotype which did not match with the enterohemorrhagic *E. coli* (EHEC) O157:H7 strain verified by the *E. coli* 0157 latex kit (Liofilchem, Teramo, Italy)**.** However, these two strains belonging to different phylogroups (D and B2), although not MDR, exhibited a large panel of virulence genes, and were moderate biofilm producers. Based on the methods of isolating the strains of *E. coli*, following direct-plating or use of an additional broth-enrichment phase before seeding on solid media, we can define two *E. coli* groups whose genotypic and phenotypic characterization showed no difference in the expression of virulence genes or MDR profile.

PCR analysis showed the presence of bacterial virulence factor genes, with *fimH* detected in all strains, followed by *kpsMT II* (11/24, 45.9%), *fyuA* (6/24; 25%) and *ibeA* (5/24, 21%). All of the isolates carried at least one virulence gene (Table 5).

Our data, taken together, establish no link between phylogeny and virulence genes in *E. coli* of equine origin. Furthermore, no correlation was found among multidrug resistance profiles, biofilm formation ability and virulence factors as shown in Table 5; whereas most of MDR strains (7/10, 70%) were moderate biofilm producers.

### 3.5. Adhesion and Invasion Assays

The 24 *E. coli* strains isolated from mares were assessed for adherence to HeLa cells. The adhesion values correspond to the number of bacteria adhered to monolayers and are summarized in Table 5. The strains recovered following direct-plating showed a lower adhesion ability than the group of *E. coli* obtained after the extra broth-enrichment phase. In fact, for this last group of *E. coli* we observed an increased binding to HeLa cells for ten *E. coli* strains, that showed a ++ value for adhesion compared to five *E. coli* strains with the same value but recovered by direct-plating. Furthermore, no strain in this group exhibited a + value of adhesion which was found for three strains recovered by direct-plating. However, these results were not statistically significant.

Interestingly, experiments with each *E. coli* strain indicated the inability of all tested strains to invade HeLa cell monolayers.

### 3.6. Hierarchical Clustering Analysis

Similarity assessment based on the genotypic and phenotypic information of isolated *E. coli* strains revealed the presence of two well-defined clusters (*p* < 0.001), as shown in Figure 1. Cluster I encompasses 20 *E. coli* strains, which were all susceptible to carbapenem (CRO), ceftiofur (EFT) and norfloxacin (NOR). Cluster II was composed of four strains, all of which were MDR.

## 4. Discussion

Mares’ infertility represents one of the major economic losses for the equine breeding industry. *E. coli*, followed by *Streptococcus* spp. and *Staphylococcus* spp., is often detected in mares with a history of repetitive infertility such as recurrent pregnancy loss [5,29]. In fact, *E. coli* is one of the bacteria most frequently isolated from mares after failure of natural or artificial insemination. Generally, the mares show a degree of physiological inflammation that resolves within 48 h of exposure to semen, whereas mares which fail to resolve their physiological post-breeding inflammation become susceptible to developing infectious endometritis [30,31,32].

In addition, the recent development and spread of MDR *E. coli* strains is of particular concern for the health of the animals as well as for the health of workers that live in close contact with horses as they may be considered putative reservoirs of MDR pathogens endowed with zoonotic potential [33].

The virulence of *E. coli* depends on the coordinated expression of several virulence factors including the ability to adhere to the host endometrium, to form a biofilm and to resist multiple classes of antimicrobials. Therefore, to better clarify the mechanisms underlying mares’ infertility, *E. coli* strains recovered from 24 mares presenting with infertility problems were analyzed. Furthermore, phylogenetic analysis was performed to determine the genetic distance among isolated *E. coli* strains. To the best of our knowledge, this is the first study that determined the phylogenetic background of *E. coli* isolated from uterine swabs of mares with fertility problems. Using this technique, we found that the majority of isolates were of phylogroup B1 and A, typical of commensal *E. coli* strains, while a few isolates were B2 and D, typical of virulent ExPEC [16,34].

Some authors have demonstrated that cattle with uterine disease are more likely to harbor group A or B1 bacteria [35]. Interestingly, among our strains obtained after the additional broth-enrichment step, the predominant phylogenetic group was D with five strains, followed by four strains belonging to the phylogenetic group B1. Instead, the phylogenetic group B1 was predominant among the strains isolated by direct-plating, followed by group A with four strains and only one strain belonging to both group D and group B2, but these differences were not statistically significant.

Antimicrobial resistance in *E. coli* has been reported worldwide and increasing rates of resistance among *E. coli* are a growing concern in both human and veterinary medicine [36]. Recently, a retrospective study of the data collected from 2006 to 2016 regarding antimicrobial susceptibility evolution in equine pathogens, showed high levels of MDR strains [37]. We observed that resistance to multiple antibiotics was detected in 41.7% (10/24) of the isolated *E. coli* strains and it was most prevalent in group B1, followed by A and D.

It is known that antimicrobials may not be effective against biofilm producing microorganisms, thus biofilm formation is considered an important virulence factor that allows bacteria to survive in the environment. Technically, biofilm is an aggregate of microorganisms surrounded by a polymeric matrix made of sugars, DNA and proteins that make bacteria highly resistant to antimicrobials [38]. Data has suggested that bacteria in either a biofilm or latent state may be responsible for many cases of subfertility. Some therapeutic approaches have been developed to try to eliminate these latent infections [39,40,41]. Moreover, as reported by different studies, approximately 80% of bacteria isolated from the equine uterus is capable of producing a biofilm [41,42,43,44]. In our study, most of the recovered *E. coli* strains were classified as moderate biofilm producers (54%), whereas a lower percentage were weak biofilm producers (29%). Furthermore, given the importance of determining the correlation between biofilm production and the profile of antimicrobial resistance, we also evaluated this aspect, and observed that the majority (70%) of the strains were MDR, and were moderate biofilm producers.

In the present study, we also found positivity for seven virulence factors in *E. coli* strains associated with equine endometritis. In particular, *fimH* was the detected factor in all *E. coli* strains, alone or in combination with at least one and up to four of the other screened virulence factors. This result agrees with other authors who described bovine intrauterine *E. coli* strains exhibiting the virulence factor gene *fimH* as the most prevalent and significant one and, consequently considered an important predictor of metritis and endometritis [45]. The same authors described *kpsMTII* in 12.8% of the *E. coli* of bovine origin [45], whilst we detected it in 46% of the isolated strains. This is a relevant finding, since recently it was suggested that *kpsMTII* has a probable key role in the progression of clinical metritis and endometritis [46]. 

In addition, the assessment of adherence to HeLa cells, found a lower adhesion ability in the strains recovered following direct-plating compared to the group of *E. coli* obtained using the additional broth-enrichment phase. It is possible to speculate that an important virulence feature belongs to the latter group of *E. coli* strains. The success of the alternative isolation protocol may be explained by the lower bacterial quantities in the uterus, thus requiring the additional enrichment-broth phase. The higher adhesion ability of these strains requires further investigation. In any case, both *E. coli* recovered by direct-plating, indicative of a high concentration in the uterus, and *E. coli* recovered only by the additional broth-enrichment phase before plating, indicative of a lower concentration, were not able to invade HeLa cell monolayers; no statistically significant differences in the expression of virulence genes or MDR profiles in the two *E. coli* groups were detected.

Taken together, we hope that these preliminary results will contribute to a better understanding of *E. coli* strains associated with fertility problems especially regarding the onset of endometritis and virulence characteristics of *E. coli* strains.

## 5. Conclusions

One of the most common challenges in equine breeding stock is endometritis, which can be difficult to resolve. *E. coli* is one of the most prevalent microbial pathogens of equine endometritis. This study characterized *E. coli* strains from mares diagnosed with reproductive disorders. The majority of them showed multidrug-resistant profiles, carried virulence genes, and were moderate biofilm producers. Results of this study indicate a predominant phylogenetic group B1, A and D and about half of them showed worrying antimicrobial multi-drug resistance profiles. Thus, further surveillance studies are required to better understand the spread of this bacterial species in mare populations. Furthermore, the role of *E. coli* and its pathological and epidemiological aspects needs further investigation, as it is necessary to provide experimental evidence regarding the possible selection or tendency of specific *E. coli* strains to be associated with endometritis in mares.

## Figures and Tables

**Figure 1 animals-13-01639-f001:**
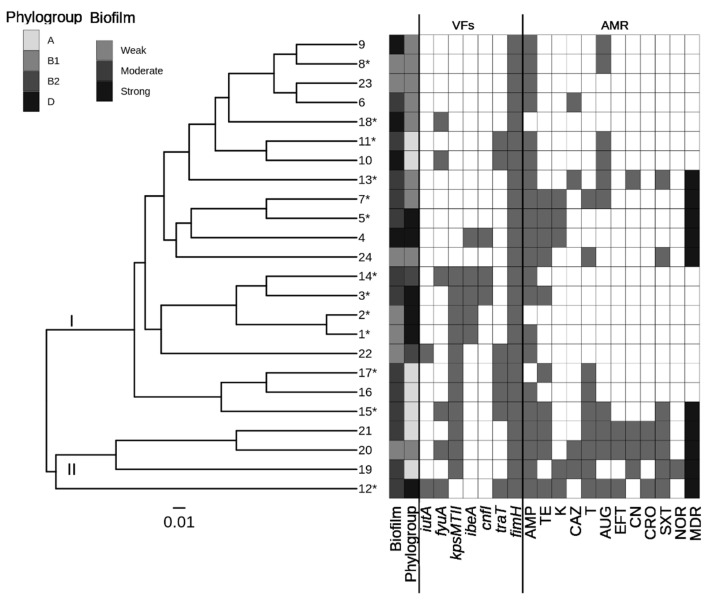
Dendrogram of the 24 *E. coli* strains based on hierarchical clustering analysis. Clusters are indicated on the tree branches (I and II) and the following features are shown: the ability to form biofilm, the phylogenetic group, the identified VF genes, the AMR profile and whether the strain is MDR. Strains that underwent a broth-enrichment step before isolation are marked with “*”.

**Table 1 animals-13-01639-t001:** Antimicrobial agents with the respective antimicrobial classes tested for *E. coli* strains.

Antimicrobial Agent	Disk Content (µg)	Antimicrobial Class	Reference
Amoxicillin–clavulanic acid (AUG)	20/10	Penicillins (alone or combined)	[19]
Ampicillin (AMP)	30		[19]
Ceftiofur (EFT)	30	Cephalosporins	[19]
Ceftriaxone (CRO)	30		[20]
Ceftazidime (CAZ)	30		[20]
Amikacin (AK)	30	Aminoglycosides	[19]
Gentamicin (CN)	10		[19]
Kanamycin (K)	30		[19]
Meropenem (MRP)	10		[20]
Enrofloxacin (ENR)	5	Fluoroquinolones	[19]
Norfloxacin (NOR)	10		[20]
Ofloxacin (OFX)	5		[20]
Sulfamethoxazole–trimethoprim (SXT)	25	Sulfonamides	[19]
Tetracycline (TE)	30	Tetracyclines	[19]
Oxytetracycline (T)	30		[19]

**Table 2 animals-13-01639-t002:** Primers used for PCR analysis of virulence genes of *E. coli* strains.

Gene	Primer Sequence (5′-3′)	Size of Product (bp)
*fimH*	TGCAGAACGGATAAGCCGTGG GCAGTCACCTGCCCTCCGGTA	508
*ibeA*	AGGCAGGTGTGCGCCGCGTAC TGGTGCTCCGGCAAACCATGC	170
*fyu*A	TGATTAACCCCGCGACGGGAA CGCAGTAGGCACGATGTTGTA	880
*iutA*	CGCAGTAGGCACGATGTTGTA CGTCGGGAACGGGTAGAATCG	300
*kpsMT II*	GCGCATTTGCTGATACTGTTG CATCCAGACGATAAGCATGAGCA	272
*traT*	GGTGTGGTGCGATGAGCACAG CACGGTTCAGCCATCCCTGAG	290
*cnf1*	AAGATGGAGTTTCCTATGCAGGAG CATTCAGAGTCCTGCCCTCATTATT	498

**Table 3 animals-13-01639-t003:** Antimicrobial resistance profiles and multidrug resistance of isolated *E. coli* strains.

Sample ID	Antimicrobial Resistance Profile	N. of Resistances to Tested Antimicrobials	MDR *
1b ^	AMP	1	-
2b	-	-	-
3b	AMP, TE	2	-
4	AMP, K, TE	3	X
5b	AMP, K, TE	3	X
6	AMP, CAZ	2	-
7b	AUG, AMP, K, TE, T	5	X
8b	AUG, AMP	2	-
9	AUG, AMP	2	-
10	AUG, AMP	2	-
11b	AUG, AMP	2	-
12b	AUG, AMP, EFT, CRO, K, SXT, TE, T	8	X
13b	AUG, AMP, CAZ, CN, SXT	5	X
14b	AMP	1	-
15b	AUG, AMP, SXT, TE, T	5	X
16	AMP, T	2	-
17b	TE, T	2	-
18b	-	-	-
19	AMP, CAZ, CN, K, NOR, SXT, T	7	X
20	AUG, AMP, CAZ, EFT, CRO, CN, SXT, TE, T	9	X
21	AUG, AMP, EFT, CRO, CN, SXT, TE, T	8	X
22	AMP	1	-
23	AMP	1	-
24	AMP, SXT, TE, T	4	X

^ b: strain isolated after broth-enrichment step. * MDR: multidrug-resistant bacterial strains. Antibiotics—AUG: amoxicillin–clavulanic acid; AMP: ampicillin; CAZ: ceftazidime; EFT: ceftiofur; CRO: ceftriaxone; CN: gentamicin; K: kanamycin; NOR: norfloxacin; SXT: sulfamethoxazole–trimethoprim; TE: tetracycline; T: oxytetracycline.

**Table 4 animals-13-01639-t004:** Biofilm formation in the 24 recovered *E. coli* strains.

Biofilm Production	% (N)
No Biofilm Producer	0% (0)
Weak Biofilm Producer	29% (7)
Moderate Biofilm Producer	54% (13)
Strong Biofilm Producer	17% (4)

**Table 5 animals-13-01639-t005:** Multidrug resistance profiles, in vitro biofilm formation, and virulence factors of *E. coli*.

Sample ID	Hem *	MDR	Biofilm Production	Phylogroup	Virulence Factors	Adhesion Values **
1b ^		-	Weak	D	*fimH*, *kpsMTII*, *ibeA*,	++
2b		-	Weak	D	*fimH*, *kpsMTII*, *ibeA*,	++
3b	X	-	Moderate	D	*fimH*, *kpsMTII*, *ibeA*, *cnf1*	++
4		X	Strong	D	*fimH*, *ibeA*, *cnf1*	++
5b		X	Moderate	D	*fimH*	+++
6		-	Moderate	B1	*fimH*	+++
7b		X	Moderate	B1	*fimH*	++
8b		-	Weak	B1	*fimH*	+++
9		-	Strong	B1	*fimH*	+++
10		-	Strong	A	*fimH*, *fyuA*, *traT*	++
11b		-	Moderate	A	*fimH*, *traT*	++
12b		X	Moderate	D	*fimH*, *fyuA*, *iutA*, *traT*	++
13b		X	Moderate	B1	*fimH*	++
14b	X	-	Moderate	B2	*fimH*, *kpsMTII*, *ibeA*, *fyuA*, *cnf1*	++
15b		X	Moderate	A	*fimH*, *kpsMTII*, *fyuA*, *traT*	++
16		-	Moderate	A	*fimH*, *kpsMTII*, *traT*	++
17b		-	Moderate	A	*fimH*, *kpsMTII*, *traT*	++
18b		-	Strong	B1	*fimH*, *fyuA*	+++
19		X	Moderate	A	*fimH*, *kpsMTII*	++
20		X	Weak	B1	*fimH*, *kpsMTII*, *fyuA*,	+++
21		X	Moderate	A	*fimH*, *kpsMTII*	+
22		-	Weak	B2	*fimH*, *kpsMTII*, *iutA*, *traT*	++
23		-	Weak	B1	*fimH*	+
24		X	Weak	B1	*fimH*	+

^ b means strain isolated after broth-enrichment step, * Hemolysis, ** The adhesion values corresponded to (+) for a value range of 1–9%; (++) between 10–19%; (+++) for values equal to or over 20%.

## Data Availability

Not applicable.

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
