# Peer review of "Genotypic and Phenotypic Characterization of Escherichia coli Isolates Recovered from the Uterus of Mares with Fertility Problems"

_animals, 2023, doi:10.3390/ani13101639_

Round 1

Reviewer 1 Report

This is an interesting study which starts to add further details to the sort of E. coli which is associated with fertility issues in horses. It adds interesting microbiological and clinical issues to the field.

I have a number of comments, mainly grammatical, which I have detailed below

Line 13- are responsible for the majority … (Reword)

Line 16- phenotypic is a typo

Line 19- invade HeLa cells as well as the carriage of …(reword)

Line 35- new insight into E. coli strains ….. (reword)

Line 37- contributing to a significant increase ….. (reword)

Line 42- affect the horse’s ability …. (Reword)

Line 48- delete ‘from

Line 55- (E. coli) have been described …(reword)

Line 61- you have ExPECs and Expc’s here- which is correct? And please define them

Line 64- is distinct correct here? Or do you mean distinguished?

Line 68- due to bacterial capability to produce biofilms….. (reword)

Line 84- delete ‘of’

Line 105- delete ‘properly’

Line 110- delete ‘of’

Line 115- Once swabs arrived at the ….(reword)

Line 118- delete precisely

Line 124- staining method, as well as catalase ….. (reword)

Line 132- all samples were ran in duplicate (reword)

Line 136- please include blood manufacturer

Line 136- were examined after up to … (reword)

Line 144- strains as susceptible …. (reword)

Table 1- is this needed? Most of it is in the text already? Perhaps only have it once?

Line 167- please have the 2 in H20 as subscript

Line 171- as non-biofilm producers…. (reword)

Line 180- agarose gel electrophoresis …(Reword)

Line 187 - delete precisely

Line 189- I am not sure that ‘were searched’ sounds correct, but I cant think of a better way to word it.

Line 189-190- please remove’ These included genes’ – as this doesn’t seem needed?

Line 197- what were the cycling conditions for the difference genes?

Line 224- is the word counted missing here after bacteria?

Table 4 - is this needed? Most of it is in the text already? Perhaps only have it once?

Line 291- delete ‘the’ before ‘D included’ ….

Line 292- strains which resulted positive….. (reword)

Line 310 delete’ the’ before’ most of the MDR strains’ ….

Line 359- susceptible to developing infectious ….(reword)

Line 377- delete ‘the’ before ‘D with 5….’

Line 381- differences resulted to be not …(reword)

Line 418-420- I am not totally sure that this makes sense- please reword

Your conclusion is a summary rather than a conclusion. Please have a look at this, and suggest what action should be taken in light of your results which is what would be expected from a conclusion.

Author Response

Reviewer 1

This is an interesting study which starts to add further details to the sort of E. coli which is associated with fertility issues in horses. It adds interesting microbiological and clinical issues to the field.

I have a number of comments, mainly grammatical, which I have detailed below

Line 13- are responsible for the majority … (Reword)

We reworded the sentence.

Line 16- phenotypic is a typo

Done.

Line 19- invade HeLa cells as well as the carriage of …(reword)

We corrected the sentence.

Line 35- new insight into E. coli strains ….. (reword)

Done.

Line 37- contributing to a significant increase ….. (reword)

Done.

Line 42- affect the horse’s ability …. (Reword)

Done.

Line 48- delete ‘from

Done.

Line 55- (E. coli) have been described …(reword)

Done.

Line 61- you have ExPECs and Expc’s here- which is correct? And please define them

We corrected it.

Line 64- is distinct correct here? Or do you mean distinguished?

We corrected it.

Line 68- due to bacterial capability to produce biofilms….. (reword)

We reworded the sentence.

Line 84- delete ‘of’

Done.

Line 105- delete ‘properly’

We did it.

Line 110- delete ‘of’

Done

Line 115- Once swabs arrived at the ….(reword)

We corrected the sentence.

Line 118- delete precisely

Done

Line 124- staining method, as well as catalase ….. (reword)

We did it.

Line 132- all samples were ran in duplicate (reword)

Thank you. We corrected as follows: all samples were run in duplicate.

Line 136- please include blood manufacturer

Done.

Line 136- were examined after up to … (reword)

Done.

Line 144- strains as susceptible …. (reword)

Done.

Table 1- is this needed? Most of it is in the text already? Perhaps only have it once?

Thank you for your advice, but we prefer maintaining the Table.

Line 167- please have the 2 in H20 as subscript

Done

Line 171- as non-biofilm producers…. (reword)

Done.

Line 180- agarose gel electrophoresis …(Reword)

Done.

Line 187 - delete precisely

Done.

Line 189- I am not sure that ‘were searched’ sounds correct, but I cant think of a better way to word it.

Thank you. We changed the verb searched with investigated.

Line 189-190- please remove’ These included genes’ – as this doesn’t seem needed?

Sorry, it was a mistake. We deleted it.

Line 197- what were the cycling conditions for the difference genes?

The sentence has been modified. As required, cycling conditions were added.

Line 224- is the word counted missing here after bacteria?

Yes, we added it.

Table 4 - is this needed? Most of it is in the text already? Perhaps only have it once?

Thank you for your advice. We prefer maintaining it in the manuscript.

Line 291- delete ‘the’ before ‘D included’ ….

Done.

Line 292- strains which resulted positive….. (reword)

Done.

Line 310 delete’ the’ before’ most of the MDR strains’ ….

Done.

Line 359- susceptible to developing infectious ….(reword)

Done.

Line 377- delete ‘the’ before ‘D with 5….’

Done.

Line 381- differences resulted to be not …(reword)

Done.

Line 418-420- I am not totally sure that this makes sense- please reword

We changed it.

Your conclusion is a summary rather than a conclusion. Please have a look at this, and suggest what action should be taken in light of your results which is what would be expected from a conclusion.

Thank you for your advice. We modified and enriched the “conclusions” section.

Reviewer 1

This is an interesting study which starts to add further details to the sort of E. coli which is associated with fertility issues in horses. It adds interesting microbiological and clinical issues to the field.

I have a number of comments, mainly grammatical, which I have detailed below

Line 13- are responsible for the majority … (Reword)

We reworded the sentence.

Line 16- phenotypic is a typo

Done.

Line 19- invade HeLa cells as well as the carriage of …(reword)

We corrected the sentence.

Line 35- new insight into E. coli strains ….. (reword)

Done.

Line 37- contributing to a significant increase ….. (reword)

Done.

Line 42- affect the horse’s ability …. (Reword)

Done.

Line 48- delete ‘from

Done.

Line 55- (E. coli) have been described …(reword)

Done.

Line 61- you have ExPECs and Expc’s here- which is correct? And please define them

We corrected it.

Line 64- is distinct correct here? Or do you mean distinguished?

We corrected it.

Line 68- due to bacterial capability to produce biofilms….. (reword)

We reworded the sentence.

Line 84- delete ‘of’

Done.

Line 105- delete ‘properly’

We did it.

Line 110- delete ‘of’

Done

Line 115- Once swabs arrived at the ….(reword)

We corrected the sentence.

Line 118- delete precisely

Done

Line 124- staining method, as well as catalase ….. (reword)

We did it.

Line 132- all samples were ran in duplicate (reword)

Thank you. We corrected as follows: all samples were run in duplicate.

Line 136- please include blood manufacturer

Done.

Line 136- were examined after up to … (reword)

Done.

Line 144- strains as susceptible …. (reword)

Done.

Table 1- is this needed? Most of it is in the text already? Perhaps only have it once?

Thank you for your advice, but we prefer maintaining the Table.

Line 167- please have the 2 in H20 as subscript

Done

Line 171- as non-biofilm producers…. (reword)

Done.

Line 180- agarose gel electrophoresis …(Reword)

Done.

Line 187 - delete precisely

Done.

Line 189- I am not sure that ‘were searched’ sounds correct, but I cant think of a better way to word it.

Thank you. We changed the verb searched with investigated.

Line 189-190- please remove’ These included genes’ – as this doesn’t seem needed?

Sorry, it was a mistake. We deleted it.

Line 197- what were the cycling conditions for the difference genes?

The sentence has been modified. As required, cycling conditions were added.

Line 224- is the word counted missing here after bacteria?

Yes, we added it.

Table 4 - is this needed? Most of it is in the text already? Perhaps only have it once?

Thank you for your advice. We prefer maintaining it in the manuscript.

Line 291- delete ‘the’ before ‘D included’ ….

Done.

Line 292- strains which resulted positive….. (reword)

Done.

Line 310 delete’ the’ before’ most of the MDR strains’ ….

Done.

Line 359- susceptible to developing infectious ….(reword)

Done.

Line 377- delete ‘the’ before ‘D with 5….’

Done.

Line 381- differences resulted to be not …(reword)

Done.

Line 418-420- I am not totally sure that this makes sense- please reword

We changed it.

Your conclusion is a summary rather than a conclusion. Please have a look at this, and suggest what action should be taken in light of your results which is what would be expected from a conclusion.

Thank you for your advice. We modified and enriched the “conclusions” section.

Reviewer 2 Report

The manuscript is well organized and well written. I recommend Accept in the present form". I have only very few minor revisions, please see the attached file. 

Author Response

Thank you for your revision.

All modified sentences and the new parts added in the revised manuscript are displayed with yellow marked changes.

Reviewer 3 Report

General comments:

The manuscript entitled "Genotypic and phenotypic characterization of E. coli isolates recovered from the uterus of mares with fertility problems" concerns on very essential problem of infertility in mares. The Authors made wide analysis of 24 isolates obtained from mares with fertility problems to extend the knowledge of this important problem in equine breeding. The work is impressive but the Authors do not avoid some shortcommings of which one is essential for better understanding the meaning of their study. The details are listed below:

Minor comments:

The species names used for the first time in the text have to be written as full names, even in case of such famous bacteria as Escherichia coli. Therefore, first using of the name should be the full mane and the next - with only initial of the genus name. It concern the use of each species name in Simple summary, Abstract and the main text.

Introduction:

line 55 - the data in brackets are not required.

Materials and Methods:

line 120 - next using of the producer name does not require to give it with the country name, Liofilchem Srl is enough. Please, verify the rest of the text to elimenate such redundant information for all producer's names. 

line 167 - H2O - the subscript shoul be used for the cipher 2.

Results:

lines 286-293 - I have serious objections to this indentation that includes data not described in Materials and Methods. First: there are three genes not included in Matherials and Methods (chuA, yjaA and TSPE4C2). Second: there is no description of phylogroups A, B1, B2 and D and the method of their identification. The third: the names of genes should be in italics. 

line 305 - the names of genes fyuA and ibeA should be in ilalics.

lines 322-325 - the sentence "It is possible...   ...was needed" is rather the part of disscussion not results.

Author Response

Reviewer 3

General comments:

The manuscript entitled "Genotypic and phenotypic characterization of E. coli isolates recovered from the uterus of mares with fertility problems" concerns on very essential problem of infertility in mares. The Authors made wide analysis of 24 isolates obtained from mares with fertility problems to extend the knowledge of this important problem in equine breeding. The work is impressive but the Authors do not avoid some shortcommings of which one is essential for better understanding the meaning of their study. The details are listed below:

Minor comments:

The species names used for the first time in the text have to be written as full names, even in case of such famous bacteria as Escherichia coli. Therefore, first using of the name should be the full mane and the next - with only initial of the genus name. It concern the use of each species name in Simple summary, Abstract and the main text.

Done

Introduction:

line 55 - the data in brackets are not required.

The data in brackets have been removed.

Materials and Methods:

line 120 - next using of the producer name does not require to give it with the country name, Liofilchem Srl is enough. Please, verify the rest of the text to elimenate such redundant information for all producer's names. 

We did not delete these information because in the authors guidelines this is required.

line 167 - H2O - the subscript shoul be used for the cipher 2.

Done.

Results:

lines 286-293 - I have serious objections to this indentation that includes data not described in Materials and Methods. First: there are three genes not included in Matherials and Methods (chuA, yjaA and TSPE4C2). Second: there is no description of phylogroups A, B1, B2 and D and the method of their identification. The third: the names of genes should be in italics. 

As requested, we modified the Materials and Methods section. Moreover, all gene names now appear in italics.

line 305 - the names of genes fyuA and ibeA should be in ilalics.

Done.

lines 322-325 - the sentence "It is possible...   ...was needed" is rather the part of disscussion not results.

We moved this sentence from results to discussion.

Round 2

Reviewer 1 Report

The manuscript reads very well, and is improved from the previous version. I have a few very very minor comments, largely just grammatical issues which I have detailed below along with suggested rewording

Line 62- act as a reservoir for human …(reword)

Line 72- biofilms conferring to microorganisms …. (Reword)

Line 392- to form a biofilm …(Reword)

Line 423- is capable of producing a biofilm … (reword)

Line 427- we observed that the majority (70% of the strains were MDR, and were moderate biofilm producers (reword)

Line 459- The majority of them showed …(reword)

Line 463- species in a mare population…(reword)

Author Response

Dear Reviewer,

Thank you very much for your comments and suggestions.

In the manuscript all the new corrections are marked in sky blue.

Line 62- act as a reservoir for human …(reword)

Thank you. We corrected it.

Line 72- biofilms conferring to microorganisms …. (Reword)

Thank you. We eliminated the comma.

Line 392- to form a biofilm …(Reword)

Thank you. We corrected it.

Line 423- is capable of producing a biofilm … (reword)

We corrected it.

Line 427- we observed that the majority (70%) of the strains were MDR, and were moderate biofilm producers (reword)

We corrected it.

Line 459- The majority of them showed …(reword)

We corrected it.

Line 463- species in a mare population…(reword)

We corrected it.
